# The Importance of Nutrition as a Lifestyle Factor in Chronic Pain Management: A Narrative Review

**DOI:** 10.3390/jcm11195950

**Published:** 2022-10-09

**Authors:** Ömer Elma, Katherine Brain, Huan-Ji Dong

**Affiliations:** 1Pain in Motion International Research Group, Department of Physiotherapy, Human Physiology and Anatomy, Faculty of Physical Education & Physiotherapy, Vrije Universiteit Brussel, 1090 Brussels, Belgium; 2Department of Physical Medicine and Physiotherapy, University Hospital Brussels, 1090 Brussels, Belgium; 3School of Health Science, College of Health, Medicine and Wellbeing, University of Newcastle, Callaghan, NSW 2308, Australia; 4Hunter Integrated Pain Service, Newcastle, NSW 2300, Australia; 5Pain and Rehabilitation Centre, Department of Health, Medicine and Caring Sciences, Linköping University, SE-581 85 Linköping, Sweden

**Keywords:** diet, nutrition, nutrition recommendation, chronic pain

## Abstract

In everyday clinical practice, healthcare professionals often meet chronic pain patients with a poor nutritional status. A poor nutritional status such as malnutrition, unhealthy dietary behaviors, and a suboptimal dietary intake can play a significant role in the occurrence, development, and prognosis of chronic pain. The relationship between nutrition and chronic pain is complex and may involve many underlying mechanisms such as oxidative stress, inflammation, and glucose metabolism. As such, pain management requires a comprehensive and interdisciplinary approach that includes nutrition. Nutrition is the top modifiable lifestyle factor for chronic non-communicable diseases including chronic pain. Optimizing one’s dietary intake and behavior needs to be considered in pain management. Thus, this narrative review reports and summarizes the existing evidence regarding (1) the nutrition-related health of people experiencing pain (2) the underlying potential mechanisms that explain the interaction between nutrition and chronic pain, and (3) the role of nutrition screening, assessment and evaluation for people experiencing pain and the scope of nutrition practice in pain management. Future directions in the nutrition and chronic pain field are also discussed.

## 1. Introduction

Chronic pain, as defined by The International Association for the Study of Pain (IASP), is pain that persists or recurs for more than 3 months [1]. Chronic pain is a serious health issue, affecting approximately 20% of adults worldwide and it is anticipated that this will continue to increase alongside the growing and ageing population [1]. There is also a significant socioeconomic burden associated with chronic pain, including high health care use and costs, high absenteeism, loss of productivity, functional impairment, and disability [2]. Due to the complexity of chronic pain and its comorbidities, both evidence and clinical practice have guided the development of integrative pain management, from monodisciplinary to multidisciplinary treatments and from multidisciplinary treatments to interdisciplinary programs, based on a biopsychosocial approach [3,4,5].

An accumulating body of evidence suggests that poor nutrition, such as malnutrition, unhealthy dietary behaviors, and a poor dietary intake can play a significant role in the occurrence, prognosis, and maintenance of chronic non-cancer pain, hereafter described as chronic pain [4,6,7]. Unhealthy dietary behaviors and a poor dietary intake is characterized by the limited intake of core nutrient-rich foods and an excessive intake of energy-dense nutrient poor foods [8]. The role of nutrition as an important lifestyle factor in pain management is gaining more attention. Over the past two decades, nutrition has occasionally been acknowledged by pain organizations, health care professionals, and consumers, and the interest in the role of nutrition in pain management has grown significantly. In a submission to the European Parliament in 2001, poor appetite and nutrition were listed, amongst others, as a burden associated with chronic pain [9]. In 2013, a qualitative study conducted by Chronic Pain Australia, an organization representing consumers experiencing pain, reported that individuals wanted more information on nutrition and pain management [10]. In 2015, Australia’s Faculty of Pain Management acknowledged that dietitians should provide input into patient care, where necessary [11]. Nutrition was also a major focus in the Consortium Pain Task Force White Paper, in 2018 [12]. More recently, in 2020, the IASP recognized the importance of optimizing one’s dietary intake in pain management strategies based on a large body of evidence, which indicated the significant effect of nutrition-based interventions on pain reduction [13,14].

Despite the growing evidence regarding the role and integration of nutrition in chronic pain management, it is still unclear how nutritional factors interact with chronic pain, the exact nature of the underlying mechanisms of this interaction, and how the nutritional care process can be implemented in chronic pain management. Thus, the aim of this narrative review is to summarize the existing evidence regarding (1) the nutrition-related health of people experiencing pain (2) the underlying potential mechanisms that explain the interaction between nutrition and chronic pain, and (3) the role of nutrition screening, assessment and evaluation for people experiencing pain, and the scope of nutrition practice in pain management.

## 2. Searching Methods and Results

### 2.1. Searching Methods

To answer the three aims of this narrative review, the existing literature was screened in an unsystematic way by three reviewers (O.E., K.B., and H.-J.D.). Each author used three online databases; PubMed, Web of Science, and Google Scholar and ensured the search was conducted no later than the 1 August 2022. Three different groups of search terms were used for each of the three study aims. Search terms can be found in Table 1. Additionally, the authors conducted forward and backward tracking of the included articles to identify studies via the reference lists or citations. Both experimental and observational studies published in English were included in the review. Studies published prior to 2000, abstracts, posters and flyers, conference proceedings, and unpublished papers were not included in the study. Studies where the full text was unavailable were also excluded.

In addition, an unsystematic search of two databases: Medline and CINAHL, as well as the European Society of Clinical Nutrition and Metabolism (ESPEN) guidelines and Practice-based Evidence in Nutrition (PEN), was conducted in July 2022 by K.B. The aim of this search was to identify the existing guidelines relating to the nutritional management of chronic pain. Search terms included the following MESH headings (“practice guideline or guideline”; “diet, food and nutrition”) and key words (“chronic pain, fibromyalgia, arthritis, back pain, musculoskeletal pain, and migraine disorders”).

### 2.2. Searching Results

Once the search on the PubMed, Web of Science, and Google Scholar databases was completed, a total of 1400 articles were identified. In total 112 eligible articles were identified and included in the paper. The full texts of two eligible studies were not available and the corresponding authors did not respond to the request for the full text [15,16]. The findings were analyzed based on the three study aims: the nutrition-related health of people experiencing pain, the underlying potential mechanisms that explain the interaction between nutrition and chronic pain, and the role of nutrition screening, assessment, and evaluation for people experiencing pain, and the scope of nutrition practice in pain management.

Following the searches on Medline and CINAHL for the guidelines related to nutrition and pain management, 112 articles were identified. None of the identified articles provided relevant information about the guidelines related to nutrition and pain. One article provided conditional recommendations and an evidence based decision aid for the use of specific dietary ingredients in chronic musculoskeletal pain [17]. However, the population of interest in this article was the United States military, which limits the generalizability of the findings to the general population. The search results from Practice-based Evidence in Nutrition (PEN) identified 32 practice guideline toolkits, of which seven corresponded to pain-related conditions such as osteoarthritis, musculoskeletal/connective tissue disorders, irritable bowel syndrome, rheumatoid arthritis, interstitial cystitis, inflammatory bowel disease, and spinal cord injury but there were no guidelines for chronic pain [18]. Of the 54 ESPEN guidelines, none were related to chronic pain [19].

## 3. The Nutrition-Related Health of People Experiencing Pain

Identifying the nutrition-related health and clinical features of people experiencing chronic pain is important for effective pain management. Malnutrition, or poor nutrition, is a health condition that occurs when an adequate nutrition cannot be acquired. An insufficient supply or consumption of nutrition (undernutrition) can lead to a person becoming underweight, while an oversupply or excessive consumption of nutrition (overnutrition) can lead to a person becoming overweight or obese. In this section, the associations between malnutrition, weight, dietary habits, and chronic pain will be discussed.

### 3.1. Underweight, Overweight, and Obesity

Population-based studies suggest that there is a higher prevalence of chronic pain amongst people with an unhealthy weight (i.e., underweight, overweight, or obese), compared to those who are of a healthy weight [20,21,22]. Underweight is defined as a body weight below the healthy weight range, while overweight and obesity are defined as an excessive and abnormal increase in white adipose tissue. The body mass index (BMI) is a weight-for-height index (kg/m^2^), which is commonly used to classify the weight status in adults. A BMI of less than 18.5 kg/m^2^ is defined as underweight, between 25.0 kg/m^2^ and 29.9 kg/m^2^ is defined as overweight and over 30 kg/m^2^ is considered obese. Obesity also consists of three subclasses: class I (30–34.9 kg/m^2^), class II (35–39.9 kg/m^2^), and class III (≥40 kg/m^2^). There are limitations associated with the BMI as it does not take into account ethnicity or body composition such as fat and muscle mass. It also does not factor in the biological, physical, economic, psychological, and social aspects that contribute to weight status. Therefore, it is important to ensure that health professionals use a holistic approach to measuring health and do not rely on weight and the BMI alone.

There is a significant focus on excessive weight and chronic pain, but it is essential to highlight that being underweight is also associated with chronic pain. The electronic Persistent Pain Outcome Collaboration (ePPOC), an Australian initiative, synthesises a standard set of data from participating chronic pain services in Australia and New Zealand. In 2020, the ePPOC reported that 2% of the 20,000 patients seeking pain management were underweight, 32% were overweight, and 39% were obese [23]. Undernutrition may also occur amongst adolescents with chronic pain and eating disorders, especially those who experience gastrointestinal issues, anxiety, and a greater functional disability [24,25]. Importantly, malnutrition and frailty are common contributors and are consequences of chronic pain. Malnutrition occurs when, over time, a person consumes too many or too few nutrients to meet their nutritional needs. This can cause adverse effects on the body, how it functions, and lead to poor health outcomes, such as a reduced life expectancy and quality of life [26]. Malnutrition may occur by not eating enough, not eating the right foods, or being unable to absorb nutrients. In Australia, up to 50% of older adults (>65 years) are malnourished or at an increased risk of malnutrition [27]. A moderate-high risk of malnutrition has also been reported by other studies conducted in different countries [28,29,30]. The evidence suggests that those experiencing hip and knee pain are at an increased risk of sarcopenia (a form of malnutrition where the loss of muscle mass occurs) and falls [31,32,33]. Pain can also impact the sensory pleasure related to food which may lead to a decrease in satiety and an increased risk of malnutrition [34]. Frailty is characterised by a decline in physical, mental, and multisystem functions and can be described as a multidimensional state of depleted physiological and psychosocial conditions [35]. A serious consequence of frailty is the increased risk of disability and death from minor external stresses, such as a mild infection or facing a stressful event [36,37]. For older adults in particular, frailty is a severe consequence related to malnutrition and chronic pain [30,38], which can predict future adverse health outcomes, such as falls and physical disability, as well as hospitalization and even mortality [36]. A recent systematic review pooled the findings from 12 cross-sectional and 12 longitudinal studies in a meta-analysis and found that older people (>60 years) with chronic pain were almost two times more likely to develop frailty after an average follow up of 5.8 years, compared to those without chronic pain [39].

The western lifestyle and diet are contributing factors impacting the global development of overnutrition, or excess weight (overweight and obesity) [40]. At the population level, overweight and obesity may explain the rising trends in chronic pain amongst middle-aged adults [41]. An Australian longitudinal study of an elderly cohort reported a relationship between fat mass, the BMI, and pain [42]. This trend has also been confirmed in the clinical populations. For example, it was found that over 25% of chronic pain patients had a comorbidity of obesity [43], much higher than the general population in Sweden [44]. The prevalence was even higher in Australia. Up to 45% of patients from a tertiary pain clinic were classed as obese [45]. Undernutrition, on the other hand, is most common among older patients [30,38,46] as well as patients with orofacial pain [47] or functional gastrointestinal disorders [48,49]. Large clinical cohort studies identified that obese patients had, in general, a worse pain profile than the normal weight patients, for example, a higher pain intensity, an increased pain interference, and more constant pain [43,50].

Recent evidence has acknowledged the importance of nutritional factors affecting specific pain conditions. For example, studies exploring fibromyalgia have demonstrated that overweight or obese patients experienced more pain, impaired function, had higher levels of depression, and medication use than patients who were normal weight [51,52]. Underweight, overweight, and obesity coexist with chronic pain due to the nutrition-related underlying mechanisms. There is an interrelationship between the nutritional status, chronic pain pathophysiology, and eating behaviors. Diet profoundly impacts the body and has a complex relationship with the pain experience [4,13]. Dietary intervention (i.e., diet patterns and eating behaviors) has also been identified as one of the integrative treatments to alleviate chronic pain [4,53]. According to the existing evidence, common chronic pain conditions have been associated with nutritional factors, such as osteoarthritis [54], rheumatic arthritis [55], fibromyalgia [56], back pain [57], irritable bowel syndrome (IBS) [49], pelvic pain (e.g., endometriosis) [58], diabetic neuropathy [59], migraine headache [60], post-herpetic neuralgia [61], and carpal tunnel syndrome [62]. A summary of common pain locations related to over- and undernutrition are shown in Figure 1. Multiple site pain conditions and spreading pain conditions, such as myofascial pain syndrome and fibromyalgia are not illustrated in the figure. Based on the IASP classification of chronic pain [63], these pain conditions may not always belong to one category (nociceptive, nociplastic, or neuropathic pain), depending on the grading of the predominant central sensitization [64,65].

Poor nutrition not only impacts chronic pain pathophysiology, but also impacts other health outcome measures. For instance, compared with non-obese patients, obese patients with chronic pain had more physical limitations [66,67], a lower psychological wellbeing [68], more sleep disturbances [69,70], a poor health-related quality of life (HRQoL) [71], and a function dependence [72,73]. Multiple physical and/or mental diseases also frequently coexist with chronic pain, such as type 2 diabetes, cardiovascular disease or metabolic syndrome, anxiety (or post-traumatic stress syndrome), and depression [21,74]. These conditions can be modified using nutrition-related treatments.

### 3.2. Eating Behaviors and Dietary Preferences in People Experiencing Chronic Pain

Optimal dietary and nutrient intake are essential elements of musculoskeletal health. In addition to weight changes, a suboptimal nutrient intake and poor eating behaviors can cause altered serum nutrient levels, which can be observed among the patients with chronic pain. For instance, high levels of serum glutamate and aspartate were reported in patients with chronic migraine, orofacial pain, fibromyalgia, and complex regional pain syndrome [75,76,77,78]. Low levels of nutrients are also commonly recognized, such as vitamin D, omega-3 polyunsaturated fatty acid, vitamin B12, magnesium, zinc, ferritin, selenium, and folic acid [56,79]. Although, these studies do not draw conclusive and direct links with the aetiology of chronic pain, it is anticipated that chronic pain patients may have altered eating behaviors, either before the onset of pain or during the development of pain.

There is also an association between a suboptimal dietary intake and some pain conditions, such as irritable bowel syndrome (IBS) and pelvic pain syndromes [80,81]. Some special but diverse dietary triggers have been reported by headache patients (particularly migraines) [60]. It is also suggested that people experiencing pain generally consume more calories, added sugars, saturated fatty acids, sodium, and caffeine. This association has been demonstrated in a cross-sectional study that found one third of males and approximately half of female participants were consuming more than the recommended daily caloric intake, moderate fat intake, and a high saturated fat intake [82]. This study also showed that the intake of vitamin D, vitamin E, and magnesium, in people experiencing chronic pain, was lower than the recommended daily intake. Data from the British Birth Cohort Study has been analyzed and showed that women with chronic pain were more likely to decrease their intake of fruit and vegetables, and increase their high fatty foods consumption over time, compared to women without chronic pain [83]. The low intake of micronutrients has also been reported in another patient population with rheumatoid arthritis [84]. Another study observed that obese osteoarthritis patients had an increased calorie, fat, and sugar intake and this impacted on their pain severity [85]. Additionally, for patients with undernutrition, pain experiences could be accompanied by a loss of appetite and a decreased food intake [24,25,86,87]. This could lead to a poor dietary intake or absorption of nutrients (i.e., medications that affect gastrointestinal functions [87]) and subsequently a decreased fat free mass and impaired physical and mental functions (i.e., daily functioning and cognitive functions [86]). 

## 4. The Underlying Potential Mechanisms That Explain the Interaction between Nutrition and Chronic Pain

The interaction between nutrition and chronic pain is bidirectional. However, it is not clear how nutritional factors interact with the pain generating mechanisms and the potential mechanisms that contribute to this relationship. Identifying and understanding these mechanisms can potentially increase the effectiveness of nutrition assessments and treatments in chronic pain management. The potential action mechanisms of the nutritional factors in chronic pain management have been identified and illustrated in Figure 2.

### 4.1. Inflammation and Oxidative Stress

Oxidative stress is defined as an increase in the reactive oxygen species produced as a byproduct of oxygen metabolism and a decrease in the ability of antioxidative compounds to detoxify cells and tissues. In addition to other factors (e.g., radiation, smoking, air pollution), dietary induced oxidative stress is one factor that can initiate and contribute to the immune cell activation and inflammation [88]. The immune cell activation, followed by a rise in oxygen consumption, also raises the amount of reactive oxygen species which can create an oxidative stress—inflammation cycle [88]. Thus, it is possible to say that the consequence of oxidative stress becomes its cause. The decreased antioxidative and detoxifying ability of the body can play a role in inflammation induced pain mechanisms [89]. Inversely, there is some evidence that an increased dietary antioxidant intake and the increased detoxifying ability of the body can alleviate pain among a chronic musculoskeletal pain population [90]. In the latest pain research, it is known that inflammation can interact with various pain mechanisms including nociceptive (pain arising due to activation of nociceptors), neuropathic (pain arising due to direct damage to the peripheral and central nervous system), and nociplastic (altered nociceptive system despite the absence of a clear nociceptive and neuropathic input) pain mechanisms [91].

Inflammation is the body’s immediate, natural, and protective response against infections and injuries. Physiologically, inflammatory processes, as a part of the immune reactions, are regulated by time. A late or prolonged inflammatory response might lose its protective effectiveness. A persistent proinflammatory state has been identified as an important risk factor for several pathophysiological conditions, including atherosclerosis, cardiovascular diseases, diabetes mellitus, obesity, cancer, and chronic pain [92]. Chronic and uncontrolled inflammation can be harmful and can lead to many acute and chronic diseases, including maintenance, occurrence, and prognosis of chronic pain [92]. This finding is supported by several pain studies on the immune system. To exemplify, immune cells evoke pain via the stimulation of nociceptors, changes in neuronal structures, and sensitization of the peripheral and central nervous systems via the release of inflammatory biomarkers [91,93].

Neuroinflammation is a localized inflammatory response that occurs in the peripheral and central nervous system [91]. In chronic pain conditions, neuroinflammation is characterized with the glial cell activation and an increased production of inflammatory biomarkers which can lead to peripheral and central nervous system sensitization [91]. Abnormal central nervous system glial cell activity has been reported in chronic pain conditions, especially nociplastic-related conditions, such as chronic non-specific low back pain, fibromyalgia, migraine, and spinal radiculopathy [94].

### 4.2. Microbiota-Gut-Brain Axis

Gut microbiota plays an important role in the human body and contributes to many structural, protective, and metabolic functions [95]. Thus, gut health constitutes an essential place in the maintenance of general health. The gut and brain have a bidirectional communication pathway and the intestinal microbiota has a modulating effect on this gut-brain axis [96]. The evidence shows that this link occurs through the connection between the vagus nerve and brainstem, via spinal afferents to the spinal cord [96]. Diversity of the gut microbiota is influenced by various factors, including medication use, mental health, infection, and nutrition which can lead to the dysregulation of the gut microbiota [97,98]. Dysregulation of the microbiota-gut-brain axis has been identified among various pathologic conditions, such as inflammatory bowel disease, diabetes, obesity, autism, depression, and chronic pain [97,98]. The accumulating evidence shows that the interrelation between nutrition and the microbiota-gut-brain axis can have a modulating effect in acute and chronic pain pathophysiology [97,98].

Microbes residing in the gut can be modified by nutritional factors. Thus, the microbiota-gut-brain axis has been identified as a target for nutritional interventions [99]. The differences in the diversity of the microbiome among the various populations that follow certain dietary patterns, such as vegetarian, vegan, and omnivorous diets, has been well documented [100]. Energy dense, unhealthy, proinflammatory dietary patterns that are nutrient poor and high in unsaturated fats, refined carbohydrates, and low in fruits and vegetables can cause a diet induced inflammation in the gut [94]. Proinflammatory cytokines released in response to unhealthy dietary patterns, activate the vagus nerve receptors located in the gastrointestinal tract. Upon activation, the vagus nerve can trigger the glial cell activation and the neuroinflammation process in the central nervous system [94]. Peripheral and central proinflammatory responses, including the aberrant glial cell activity, contribute to the maintenance, occurrence, and prognosis of chronic pain [94]. Targeting the gut microbiota with nutritional interventions in chronic pain populations is a promising approach for pain management.

### 4.3. Disturbed Glucose Metabolism

Diabetes has been reported as an important risk factor for chronic pain. In addition to neuropathic pain, chronic non-neuropathic pain conditions, such as fibromyalgia, chronic wide-spread pain, chronic low back, and neck pain are more common among people with diabetes, compared to people without diabetes, especially amongst those who have poorly controlled diabetes [101,102]. Patients with chronic musculoskeletal pain have been identified as having a disturbed glucose metabolism, including an increased fasting glucose level, an increased insulin resistance, a higher postprandial glycemic response, and a higher prevalence of type-2 diabetes mellitus [103]. A well-known antihyperglycemic medicine, metformin, which is commonly used to treat type-2 diabetes has also shown it can significantly alleviate pain in chronic pain populations and thus could be a potential treatment for people experiencing chronic pain [104].

An excessive carbohydrate intake and a decrease in glucose metabolism efficiency can increase reactive oxygen species and evoke an oxidative stress response [105]. The oxidative stress response is an endogenous factor that can activate toll like receptors and initiate inflammatory reactions in the peripheral and central nervous systems [105]. Thus, the identification of a disrupted glucose metabolism and targeting glucose regulation constitute significant places in chronic pain management [105]. Studies exploring the effect of low-carbohydrate diets, including the ketogenic diet, have revealed promising results including improvements in the glucose metabolism [106]. In another study, people with chronic musculoskeletal pain, who followed a low carbohydrate diet, had a decrease in serum inflammatory biomarkers and pain sensitivity [107]. Studies that explored the action mechanism of a ketogenic diet on chronic pain suggested that the carbohydrate intake played a role in neuroinflammation and central sensitization [108]. However, it is also important to consider the weight reducing effect of low-carbohydrate diets. A decrease in adipose tissue may also improve pain sensitivity in chronic pain populations and therefore, obesity requires special attention in terms of its role in the interaction between nutrition and pain generating mechanisms [6,109].

### 4.4. Disrupted Lipid Metabolism

Lipids are essential for several bodily functions, and are one of the body’s main energy sources. Nutrition strategies, including the modification of single nutrients, supplements, or overall eating patterns, can affect serum lipid profiles in both positive and negative ways. To exemplify, an excessive intake of saturated fats, dietary fructose, and an overall western style of eating increases low-density lipoprotein (LDL), triglycerides, and decreases high-density lipoprotein (HDL). A high LDL cholesterol level increases the risk of cardiovascular disease (CVD) and is commonly known as “bad cholesterol”, while a high HDL cholesterol level is protective, reduces the risk of CVD, and is often referred to as “good cholesterol”. Omega-3 unsaturated fatty acids, antioxidants, intermittent fasting, and adherence to the Mediterranean diet, can have the reverse effects on the same lipid biomarkers [110,111,112,113,114].

A disrupted lipid metabolism also plays a role in various health conditions, such as atherosclerosis, diabetes, cardiovascular diseases, metabolic syndrome, and obesity. [115]. The role of a disrupted lipid metabolism in chronic pain is gaining more attention and targeting this mechanism via dietary factors is a promising approach for chronic pain management. For instance, low back pain has been found to be prevalent among individuals with a decreased lumbar blood supply [116]. The relationship between the decreased lumbar blood supply and spinal pain constitutes a base for the atherosclerosis theory of the persistent non-specific low back pain. Prevalence of low back pain has been found inversely associated with the serum HDL cholesterol and positively associated with serum triglycerides and LDL cholesterol, which overall contribute to the atherosclerosis hypothesis [116,117]. Additionally, compared to healthy controls, fibromyalgia patients have shown a disrupted serum lipid profile and this disruption was found to be positively associated with pain sensitivity [118]. In a systematic review, the biomarkers of the serum lipid metabolism, including the decreased serum HDL cholesterol, the increased serum LDL cholesterol, and triglycerides, was found to be strongly associated with musculoskeletal pain arising from tendinopathy [119].

### 4.5. Obesity/Overweight

Obesity is associated with a proinflammatory state and is an important risk factor for various metabolic changes and chronic diseases, including cardiovascular diseases, cancer, diabetes mellitus, and chronic pain [109,120,121]. The existing evidence suggests that there is a concurrence and bidirectional relationship between obesity and chronic pain [109]. Obesity has been associated with several chronic musculoskeletal pain conditions including osteoarthritis, fibromyalgia, pelvic pain, and chronic low back pain [122]. It has been hypothesized that overweight/obesity play an important role in chronic pain by two main mechanisms; first increasing the mechanical load on neuromusculoskeletal structures and second, initiating or contributing to neuroimmune reactions, namely chronic low grade systemic inflammation [123].

Increased adipocytes and adipose tissue are positively associated with increased macrophages and promote inflammatory responses such as an increase in inflammatory cytokines (IL-6, TNF-alpha) and acute phase proteins (CRP) [124]. Excessive adipose tissue also increases the relocation of inflammatory cytokines into the central nervous system and promotes the activation of glial cells which can eventually play a role in nociplastic pain [125].

Exposure to high saturated fat and energy dense dietary patterns increase the circulated inflammatory cytokine levels. An in vivo study using an animal model suggests that exposure to a diet rich in saturated fat for one day causes the glial cell activation for two weeks in rats [126]. Dietary patterns that restrict the caloric intake have been shown to relieve pain in people with chronic musculoskeletal pain [127].

### 4.6. Epigenetic Factors

Epigenetics can be explained as a change in the gene expression without any change in the deoxyribonucleic acid (DNA) sequence. Epigenetic mechanisms are divided into three main categories, namely DNA methylation, histone modifications, and non-coding ribonucleic acid (RNA) interference [128]. Almost every cell in the body has the same DNA. However, each cell has different activated or highlighted genes in the DNA. Epigenetics explains the interaction between nature (genes) and nurture (environment), and how the genes we inherit interact with environmental factors including diet [128]. Many epigenetic mechanisms are reversible and modifiable which make them an attractive therapeutic target.

Nutrition is a major modifiable lifestyle factor that has the ability to alter the epigenetic regulation and can cause an epigenetic dysregulation. Additionally, epigenetic markers also have the ability to alter the body’s response to certain dietary intake and patterns [129,130]. Dysregulation of the epigenetic markers can alter the gene expression, protein synthesis, cell function, and metabolism and can lead to chronic diseases [131,132].

Recent findings show that epigenetic changes can alter the expression of nociceptive or antinociceptive genes [133]. Moreover, the epigenetic dysregulation can play a role in the transition from acute to chronic pain [134]. Preclinical studies have shown an increase in inflammatory responses after the consumption of a diet rich in saturated fats and a high-carbohydrate diet via DNA methylation [135]. The DNA methylation level of genes that promote inflammatory cytokines, especially TNF-alpha, has been associated with obesity and an omega-6 polyunsaturated fatty acids intake [136]. Saturated fatty acids are known for their inflammatory characteristics and a higher intake of saturated fatty acids has been associated with the DNA methylation level of the genes that play an essential role in the inflammatory biomarker synthesis and insulin resistance [137]. Alternatively, nutrients and foods with anti-inflammatory properties, such as omega-3 polyunsaturated fatty acids, extra virgin olive oil, curcumin, and polyphenols showed anti-inflammatory effects via its effects on the DNA methylation processes in immune cells [138]. Early findings show that there is an interaction between nutrition and the epigenetic factors and this has an important role in chronic pain and the associated mechanisms, such as obesity, a disturbed glucose metabolism, and gut microbiota diversity. Although the use of genetic and epigenetic data in chronic pain management is still in a very early phase, the potential for the development of personalized pain medicine, or precision pain medicine is both promising and innovative [139].

## 5. Implementation and Scope of Nutrition in Chronic Pain Management

The relationship between nutrition and chronic pain is important and complex, yet traditionally, nutrition has been underrepresented in the evidence-based biopsychosocial and lifestyle approach to pain management. Pain management is multifaceted and must include an interdisciplinary approach. As such, health professionals need to be aware of, and be able to identify nutrition-related risk factors associated with chronic pain, provide basic nutrition-based treatment strategies, and know when and how to refer to a dietitian for more complex issues and advice. A thorough nutrition assessment and treatment plan should be included in all pain management programs. Dietitians can be a valuable part of a multidisciplinary team and can provide comprehensive assessments and treatments.

### 5.1. Nutrition Assessment for Chronic Pain

There is growing evidence to show that there is an association between diet and health outcomes that are important for people experiencing chronic pain. Therefore, a nutrition assessment should be conducted early in treatment. This may be through a brief, opportunistic intervention that a health professional (e.g., general practitioner (GP), nurse or allied health professional) may provide to a patient, a structured nutrition screening process at a pain clinic, or a comprehensive dietary assessment conducted by a dietitian. There are several nutrition-related risk factors associated with chronic pain and these should be addressed in a dietary assessment. The factors include malnutrition, weight change, the presence of other comorbidities, abnormal biochemistry results, appetite or gastrointestinal complaints, and a poor dietary intake.

#### 5.1.1. Malnutrition Screening

Malnutrition screening is a vital component to consider when conducting a nutrition assessment for people experiencing pain. It is essential for those at an increased risk of malnutrition, such as older adults, those with orofacial pain or functional gastrointestinal conditions [30,38,46,47,48,49]. The process should include the use of a validated malnutrition screening tool. There are several validated malnutrition screening tools, such as the Malnutrition Universal Screening Tool, Malnutrition Screening Tool, Mini Nutritional Assessment-Short Form, and the Nutrition Risk Screening Tool [140,141]. These tools use similar parameters and are reliable in identifying people who are malnourished or at risk of malnutrition. Most health services or facilities use a specific screening tool based on their population, the complexity, and sensitivity of the tool. This serves as guide for health professionals when choosing the tool. People who fall into the malnutrition or in the at risk of malnutrition categories, should be referred to a dietitian.

#### 5.1.2. Monitoring Weight Changes

Measuring weight can be confronting for patients and given that the BMI is not always an accurate measurement of weight, it is important to discuss weight measurements with patients to ensure they are comfortable. Monitoring changes in weight over time, (i.e., monthly) can be a useful indication of under- or overnutrition. It can also be useful in identifying serious illnesses associated with sudden and unplanned weight loss, such as cancer [142] and inflammatory bowel disease [143]. The BMI or a waist circumference can help to identify changes in weight over time. However, as previously stated, the BMI must be used with caution. A visual assessment or asking a patient if their clothes are tighter or looser can be less confronting and still obtain the same information. Dietary strategies for pain management are likely to result in improvements in overall health and potentially weight loss. For a successful and sustained nutrition-related change, focusing on pain is more likely to resonate with a patient, compared to weight loss [144]. Patients are more likely to feel validated and motivated which will assist with behavior changes [144,145].

#### 5.1.3. Identifying Other Comorbidities

Several studies have shown that people experiencing chronic pain also have multiple comorbidities [45,146]. Many of these comorbidities can be influenced by nutrition, such as cardiovascular disease (CVD), diabetes, and depression [147,148,149]. Recent studies have found that people with musculoskeletal pain were twice as likely to have CVD than those without [150], people with diabetes were 1.4 times more likely to report lower back pain and 1.2 times more likely to report neck pain [102], and people with depression were three times more likely to experience non-neuropathic pain and six times more likely to experience neuropathic pain [151]. Globally, a poor diet is the top modifiable risk factor for morbidity [152]. Chronic pain and chronic health conditions share a relationship with inflammation, oxidative stress and a poor diet quality [153]. As such, many of the nutrition recommendations in Table 2 may not only improve pain experiences but may also improve the severity and impact of other chronic health conditions [53]. A referral to a dietitian should also be considered so a detailed and tailored assessment and relevant advice can be provided. 

#### 5.1.4. Identifying Abnormal Biochemistry Results

As outlined in Section 3.2, there are several micronutrient deficiencies that are commonly associated with chronic pain, such as the B-group vitamins and Vitamin D [56,79,82,157]. These can be identified through routine pathology tests. Dietitians can also identify these deficiencies through comprehensive dietary assessment methods. While it is unclear what the exact relationship is between chronic pain and micronutrient deficiencies, evidence suggests that some vitamins, especially the B-group vitamins, play a role in maintaining the health of the nervous system and pain-signaling pathways [121,158]. Additionally, this paper also outlines several underlying mechanisms associated with chronic pain, including a disrupted lipid and glucose metabolism. Abnormal serum lipids, glucose, and insulin can be used to identify issues with metabolism that may be present and contributing to pain experiences in people with pain [94].

#### 5.1.5. Identifying Gastrointestinal Complaints

Gastrointestinal complaints are common in people experiencing chronic pain. A recent systematic review found that people with irritable bowel syndrome were 1.8 times more likely to have fibromyalgia and that 50% of those with fibromyalgia had a least one functional gastrointestinal disorder [159]. Functional gastrointestinal disorders (FGIDs) comprise a variety of chronic and recurrent gastrointestinal symptoms that cannot be explained by structure or biochemical abnormalities [160]. While the exact nature of FGIDs is still unclear, it has been linked with an altered gut-brain communication and a hypersensitivity of the enteric nervous system [161]. People experiencing chronic pain should be screened for symptoms associated with FGIDs, such as abdominal pain, dysphagia, dyspepsia, diarrhea, constipation, and bloating [161]. There are several strategies (e.g., medication, exercise, cognitive behavior therapy, and nutritional strategies) that are used to manage symptoms using an interdisciplinary approach. From a nutrition perspective, there are a variety of options. These include a diet that is low in fermentable oligosaccharides, disaccharides, monosaccharides and polyols (FODMAPs), modifying the fiber intake, or restricting certain foods, such as caffeine, alcohol, spicy foods, and foods high in fat [162,163]. Given the variety in dietary strategies, it is important to refer patients to a dietitian to ensure a comprehensive dietary assessment is undertaken before trialing these strategies. Some of these dietary strategies can result in an inadequate nutrient intake. For example, a low FODMAP diet is an elimination diet and removing foods and food groups from the diet leads to a nutritional inadequacy [164]. Thus, it is vital that dietitians work with patients to ensure they can meet their nutritional needs while trialing these strategies.

#### 5.1.6. Assessing the Dietary Intake

A poor dietary intake is another risk factor for chronic pain. Many people experiencing pain are likely to have a limited intake of the core nutrient-rich foods and an excessive intake of energy-dense nutrient-poor foods [82,83]. Health professionals can measure the dietary intake by assessing the diet quality. Diet quality can be defined as a varied nutritious diet, which provides individuals with adequate amounts of essential nutrients needed to support overall health and wellbeing [165]. Optimizing the diet quality will address several risk factors in one strategy. Diet quality can be measured using a diet quality index or diet score such as the Diet Quality Index-International [166], the Healthy Eating Index [167] or the Dietary Inflammatory Index [168]. Some can be automatically calculated, such as the Australian Recommended Food Score (ARFS), which can be determined by completing an online questionnaire called the Healthy Eating Quiz [169]. Every country also has a set of dietary guidelines and health professionals can compare a patient’s intake against these guidelines to determine areas for improvement. However, as acknowledged in Philpott (2019), chronic pain services would significantly benefit from including dietitians and their skills in the assessment, medication, and support of diets specific to chronic pain [170]. Dietitians can conduct detailed and tailored dietary assessments which provide more insight into a patient’s dietary intake and can identify more areas for improvement.

### 5.2. Nutrition Treatments for Chronic Pain

Evidence suggests that following a predominately plant-based eating pattern (e.g., vegetarian, vegan, or flexitarian eating pattern) or a Mediterranean eating pattern (characterized by a high consumption of fruit, vegetables, legumes, wholegrains, dairy, olive oil, moderate consumption of fish, and small amounts of red meat) or an optimizing diet quality are most effective at reducing pain experiences [53]. The evidence available in the scientific literature is also supported by practice guidance toolkits [18]. However, these guidelines are limited to specific chronic pain conditions such as osteoarthritis, rheumatoid arthritis, and fibromyalgia. These toolkits also recommend predominately plant-based eating patterns, healthy fats and oils, and consuming a wide variety of nutritious foods. The evidence presented in the literature and toolkits can be synthesized into dietary recommendations that health professionals can provide to people experiencing pain (Table 2).

The evidence also indicates that reducing and limiting the intake of added sugar and energy-dense, nutrient poor, or ultra-processed foods will reduce the underlying mechanisms such as inflammation and oxidative stress that contribute to chronic pain experiences [94,171]. Ultra-processed foods undergo several industrial food processes and contain high amounts of sugar, chemically modified protein (e.g., hydrolyzed proteins), oil products (e.g., hydrogenated oils), and food additives [172]. They also contribute to a poor diet quality, metabolic health, and the development of chronic health conditions [173]. Examples of these foods include soft drinks, sweet or savory packaged snacks, and processed meats. National dietary guidelines recommend limiting the consumption of these foods, both in the amount consumed and in the frequency of consumption. In addition, the World Health Organization (WHO) recommends that adults limit their added sugar intake to less than 10% of their total caloric intake [174]. This includes foods such as table sugar, syrups, sweet packaged snacks and baked products, and sugar-sweetened beverages.

It is also important to consider the barriers or practical implications to adhering to a particular eating pattern. These include: ability and access to shop, prepare and cook food, pain flare-ups, cost, culinary skills, sleep, gastrointestinal symptoms, food intolerances, environment, motivation, and mood [53,175]. As part of a multidisciplinary team, a dietitian can work with the patient and their health care team to develop a sustainable plan that improves pain experiences, other health outcomes, and that can be adhered to over a long period of time [175].

Social determinants of health, such as education, socioeconomic status, access and quality of essential services, and the social environment, also play a role in an individual’s ability to access nutritious and affordable food. Food insecurity is the inability to reliably access adequate and affordable nutritious food and it is associated with chronic pain and poor mental health. Findings from a recent survey of 200 adult food bank users in the United States, found that 53% of respondents reported experiencing chronic pain [176]. In this study, after controlling for age and gender, depression, and chronic pain significantly predicted food insecurity. A study which analyzed data from approximately 80,000 Canadians aged ≥12 years found that those who were food-insecure were 1.3 times more likely to experience chronic pain and almost 2.7 times more likely to have used prescription opioids in the last year [177]. This demonstrates that multidisciplinary teams must explore barriers, practical implications, and social determinants of health when it comes to nutrition and pain.

Other health professionals, such as psychologists, occupational therapists, and physiotherapists can also provide valuable advice and guidance that will work, in combination with the advice and guidance provided by the dietitian, to address some of these practical implications. For example, a psychologist can help address mood and motivation, an occupational therapist can undertake a functional assessment and provide advice on how to participate in nutrition and food-related activities, such as cooking, and physiotherapists can assist by facilitating people to build their strength and mobility which will help with accessing food.

A common denominator for all health professionals is behavior change. These practical implications can also be considered barriers that may make behavior change difficult. Behavior change is a fundamental part of the biopsychosocial and lifestyle approaches to pain management. Models and frameworks, such as the Behavior Change Model [178], can be used to understand and implement behavior change to overcome these barriers. It is important that all health professionals in a multidisciplinary team are familiar with behavior change models and incorporate behavior change techniques in their practice.

### 5.3. Scope of Practice

Dietary changes vary in their simplicity and sustainability. Some changes are easy, and others are harder to implement and sustain over time. These changes can be categorized into general healthy eating, basic, or complex recommendations for chronic pain, and personalized medical nutrition therapy as outlined in Figure 3.

In a multidisciplinary team, all health professionals should understand all of the components involved in pain management, including nutrition. However, it must be acknowledged that all health professionals have a particular area of expertise. Dietitians are experts qualified to provide medical nutrition therapy using the nutrition care process. In the nutrition and chronic pain scope of practice, all health professionals, should understand general healthy eating and have a basic understanding of nutrition-related recommendations for chronic pain. Pain management teams include medical, nursing, physiotherapy, psychology, and other allied health professionals and all have a significant role in providing relevant and appropriate health education to patients, including nutritional recommendations. However, a comprehensive understanding of nutrition-related recommendations for chronic pain and personalized medical nutrition therapy, should be provided by credentialed dietitians (e.g., Accredited Practising Dietitian or Registered Dietitian) who have undertaken approved study at university and registered with their respective national dietetic association (e.g., Dietitian’s Australia or British Dietetic Association). Regardless of a patient’s needs, whether it be advice on general healthy eating, basic or complex nutrition recommendations, or personalized medical nutrition therapy, a dietitian can provide valuable input at all stages.

#### 5.3.1. General Healthy Eating

Each country has dietary guidelines for healthy eating. Dietary guidelines promote healthy eating and lifestyle behaviors, rather than treating nutrition-related diseases. They convey the big picture and encourage the consumption of a variety of nutrient-dense foods. While this is not specific to chronic pain, many people do not meet the recommendations in these guidelines, which will impact on their overall health and wellbeing. For example, in Australia, only 6% of adults met the recommended daily amount of fruit and vegetables in 2020–2021 [179]. This highlights that health professionals still need to support people to improve their dietary intake to align with the recommendations.

#### 5.3.2. Basic Nutrition Recommendations for Chronic Pain

Basic recommendations for chronic pain, such as those provided in Table 2, are simple recommendations that all health professionals can support and help their clients to achieve. These are more specific to chronic pain as they address the underlying mechanisms, such as inflammation and oxidative stress.

#### 5.3.3. Complex Recommendations for Chronic Pain

Some people have a more complicated relationship with nutrition and pain, this is often due to the multiple barriers and/or underlying mechanisms and/or comorbidities, such as FGIDs. As previously mentioned, FGIDs are often associated with chronic pain and nutrition-related strategies should be provided by a dietitian.

#### 5.3.4. Personalized Medical Nutrition Therapy

Dietitians are trained to provide personalized medical nutrition therapy which acknowledges that a one-size fits all approach is not appropriate as individuals have different circumstances. Using the Nutrition Care Process [180], dietitians translate evidence-based nutrition information into tailored and practical dietary advice. Dietitians also participate in ongoing professional development to keep apprised of new or updated information. Patients who want to trial an elimination diet must do so with the support of a dietitian to ensure they maintain an adequate nutrition. Patients who have multiple comorbidities requiring multiple nutrition strategies should see a dietitian who can work with them to facilitate the appropriate dietary changes.

## 6. Future Perspectives

Nutrition interventions deserve to be an essential part of pain management [4,14]. Diet is a modifiable lifestyle factor that can be improved through nutrition interventions. At present, although nutrition is gaining more attention in pain management, current evidence mostly comes from preclinical studies, observational trials, or experimental studies that lack control groups or long-term follow up periods. Most of the available human trials are observational studies and explore the association between nutrition and pain, but do not clarify the causality behind the interactions between nutritional factors and pain. Future trials should consist of high-quality randomized controlled trials in more specific populations and on various chronic pain conditions. Studies in clinical settings need to be carefully designed to match patient characteristics due to the complexity of chronic pain. Clinical trials may explore both pain and nutritional-related comorbidities, such as patients with excess weight and vulnerable groups with somatic (i.e., frail elderly with multimorbidity) or psychiatric diseases (i.e., eating disorders). Additionally, it is of interest to explore more specific dietary patterns and dietary quality in the clinical populations so that real-world data may support the evidence of the appropriate dietary therapies in target patient populations. The latest research, however, is usually based on the general populations [57,181].

Current dietary guidelines provide advice for the general population to ensure people consume adequate nutrition and prevent chronic diseases [8]. However, as suggested in the observational studies, the needs of people experiencing pain differ from those who do not experience chronic pain. Thus, specific dietary guidelines need to be developed for chronic pain. These guidelines need to take into consideration the specific needs of people experiencing pain. They also need to be incorporated into the assessment and diagnosis procedures for chronic pain. Nutritional screening and assessments should be specific for people experiencing pain and be adapted, based on the evidence related to the pathophysiology and underlying mechanisms of chronic pain. The lack of clinical guidelines for nutrition and chronic pain indicates that the evidence needs to be synthesized into a clinical guideline for nutrition and chronic pain.

Investigating the relationship between nutritional factors and the physiological processes of the body are highly complex due to the difficulty of isolating the impacts of nutritional factors from the high number of confounding factors among individuals. As a starting point, developing general dietary guidelines for specific populations and subgroups constitutes an important place in pain medicine. However, based on novel and innovative technology and science, there will most likely be a shift from a “one size fits all” approach to personalized nutritional (pain) medicine. Improving technological development will allow researchers and clinicians to deal with large and more complex amounts of data which can be adapted to pain and nutrition. To exemplify, artificial intelligence and machine learning based software and applications have a great potential to collect real life and complex data from individuals and to capture meaningful insights for research and clinical purposes.

It is also important to explore the barriers people with chronic pain experience when adopting healthier eating patterns, to ensure successful and meaningful change. One perspective is to address other lifestyle factors in parallel to nutritional intervention. Strong associations were found between dietary habits, sedentary behaviors, and physical activity, especially among younger people [182,183]. Diet and sleep also have a bidirectional relationship [184]. Since sedentary behavior and sleep disturbances are extremely common among patients with chronic pain, future nutrition-based studies may consider the evaluation and combined effects of lifestyle interventions (physical activity, sedentary behavior, sleep, and dietary therapy).

## 7. Conclusions

The relationship between nutrition and chronic pain is complex but traditionally underrepresented despite the emerging evidence which indicates that poor nutrition and dietary intake may play a key role in the development and management of chronic pain [13,53]. This paper highlights that nutrition contributes to chronic pain patients’ profiles; there is a strong link between the underlying mechanisms of chronic pain and nutrition and there is a place for a nutrition-related assessment and management in chronic pain management. Health professionals and chronic pain services need to be aware and understand the role nutrition plays in chronic pain management. With this growing evidence base, nutrition assessments and management plans should be incorporated into the care of people experiencing chronic pain.

## Figures and Tables

**Figure 1 jcm-11-05950-f001:**
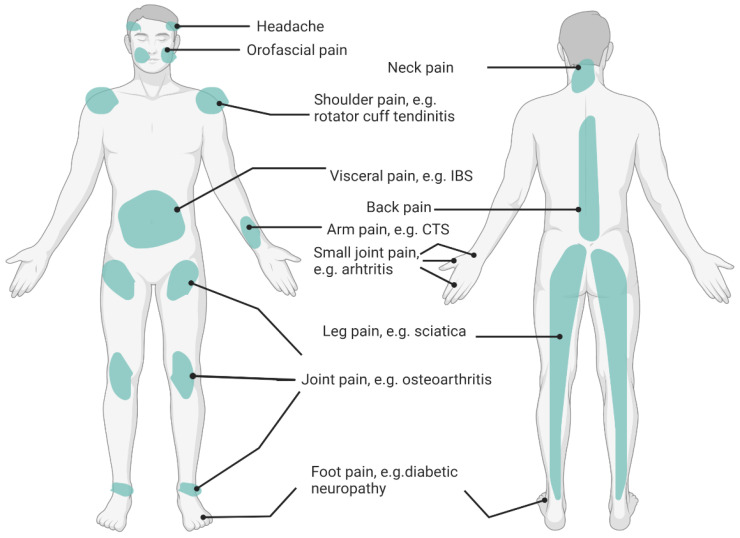
Pain Sites Related to Poor Nutrition. IBS: Irritable Bowel Syndrome; CTS: Carpal Tunnel Syndrome.

**Figure 2 jcm-11-05950-f002:**
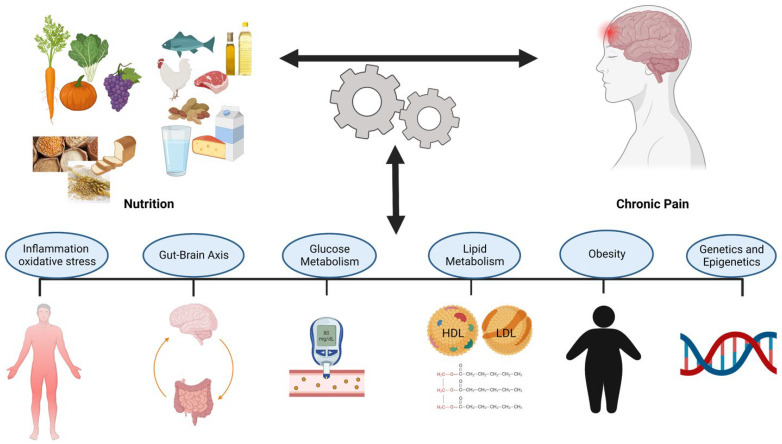
Potential Mechanisms of the Interaction Between Nutrition and Chronic Pain.

**Figure 3 jcm-11-05950-f003:**
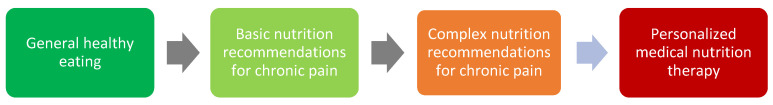
Nutrition and the chronic pain scope of practice.

**Table 1 jcm-11-05950-t001:** Search Terms.

Search Terms for the “The Nutrition-Related Health of People Experiencing Pain”	
**Pain**	“Chronic Pain”; “Myalgia”; “Fibromyalgia”; “Arthritis”; “Osteoarthritis”; “Headache”; “Migraine”
**Nutrition**	“Diet”; “Dietary Pattern”; “Eating Behavior”; “Nutrition”; “Malnutrition”; “Underweight”; “Obesity”; “Overweight”; “Fat Mass”
**Search terms for the “The underlying potential mechanisms that explain the interaction between nutrition and chronic pain”**	
**Pain**	“Chronic Pain”; “Myalgia”; “Fibromyalgia”; “Arthritis”; “Osteoarthritis”; “Headache”; “Migraine”
**Nutrition**	“Diet”; “Dietary Pattern”; “Eating Behavior”; “Nutrition”; “Obesity”; “Overweight”; “Fat Mass”
**Mechanism**	“Metabolism”; “Inflammation”; “Oxidative Stress”; ”*genetics”
**Search terms for the “The role of nutrition screening, assessment, and evaluation of chronic pain patients and the scope of nutrition practice in the pain management process**	
**Pain**	“Chronic Pain”; “Myalgia”; “Fibromyalgia”; “Arthritis”; “Osteoarthritis”; “Headache”; “Migraine”
**Nutrition**	“Diet”; “Nutrition”; “Food”; “Dietary Pattern”; “Eating Behavior”; “Dietary Assessment”; “Gastrointestinal Symptoms”

* Wildcard represents unknown characters and identify word combinations by filling in automatically.

**Table 2 jcm-11-05950-t002:** Nutrition recommendations for people experiencing chronic pain.

Food Group/Nutrient	Recommendation	Rationale	Practical Tips
**Fruit and vegetables**	Encourage the consumption of fruit and vegetables. Aim for a variety and wide range of bright colors.	Fruit and vegetables contain phytonutrients which reduce oxidative stress and inflammation.	Choose frozen fruits and vegetables options to reduce preparation time and effort, food waste, and increase variety. Nutrients are retained through freezing.
**Breads, cereals, and grains**	Choose wholegrain and fiber-rich options. Aim for foods with a low glycemic index. *	Provides slow but sustained energy. Fiber & prebiotics—improves gut health and feeds the gutmicrobiome which may play a role in pain and inflammation.	Swap bread, pasta, and rice for wholegrain options. Swap high GI foods for low GI options.
**Meat and meat alternatives**	Choose lean meats (e.g., chicken, fish, and small amounts of red meat). Prioritize oily fish, legumes, nuts, and seeds.	Contain healthy fats which reduce inflammation. Build strength to address deconditioning associated with chronic pain.	Swap processed meats for lean meats. Choose tinned fish and legumes to save time and effort with meal preparation.
**Dairy and dairy alternatives**	Choose high quality dairy foods (e.g., milk, cheese, and yoghurt).	Contains protein to build strength, variety of fats, and important vitamins and minerals.	Choose reduced fat options where possible. Pre-sliced or grated cheese will reduce energy and time needed to prepare meals. Individual tubs of natural or Greek yoghurt (no added sugar) are an easy snack
**Healthy fats and oils**	Omega-3 and monounsaturated fats.	Reduces inflammation.	Swap cooking oil for olive or canola oil.
**Drinks**	Consume 2–3 L water/day. Limit caffeine.	Dehydration increases sensitivity to pain [154,155,156].	Carry a water bottle with you and set a goal to consume it all within a set time period.
**Added sugar and ultra-processed food**	Reduce and limit intake.	Increases inflammation and oxidative stress.	Swap sugar-sweetened beverages and energy drinks for mineral water. Choose healthy snack options, e.g., fruit, nuts, wholegrain crackers, and cheese or popcorn. Utilize minimally processed foods to facilitate home cooking rather than convenience/takeaway options, e.g., pre-cut vegetables, tinned fish and legumes, tomato based sauces, and microwave rice.

* Glycemic index is a ranking system for carbohydrate foods and is based on the speed of digestion and impact on the blood glucose levels over a period of time. Glucose has a GI of 100 and this is the reference used for other foods. Carbohydrates that breakdown quickly and lead to a sharp increase in blood glucose levels are high GI foods. Carbohydrates that breakdown slowly and lead to a gradual and sustained increase of the blood glucose levels are low GI foods.

## Data Availability

Not applicable.

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
