# Peer review of "The Importance of Nutrition as a Lifestyle Factor in Chronic Pain Management: A Narrative Review"

_jcm, 2022, doi:10.3390/jcm11195950_

Round 1
Reviewer 1 Report
Thank you for allowing me to review this article. Your narrative reviewed aimed to answer the question: Among adults with chronic, non-cancer pain, what underlying mechanisms drive relationships between chronic pain symptoms and nutritional factors, and what nutritional care processes can target these mechanisms?
1. The authors report that this was an unsystematic review of the literature, and yet total number of articles identified, screened, and included was not mentioned. Please provide this evidence to the reader to ensure breadth of the search. Also, the article does not mention whether the search was limited by year of publication. Please clarify if this is the case
2. Recent literature suggests a link between food insecurity and chronic pain symptoms and this was not cited in the paper. Mentioning this link might be important for readers, especially when attempting to recommend nutritional interventions to manage pain.
3. The authors do not cite Registered Nurses as potential health partners with patients suffering chronic pain. Nurses may provide critical health promotion education including nutritional recommendations to patients across the continuity of care and should therefore be addressed as well.
4. Under 5.3, scope of practice, the authors mention "qualified dieticians" can provide personalized medical nutrition therapy (note that nutrition is misspelled in Figure 3, final box). Please provide a definition, such as a specialty certification or formal training course that dieticians (or other health professionals) can pursue to become expert in this domain.
5. Line 577: please spell out FGID in this section before use
Reviewer 2 Report
Good job with this article. The topic was interesting. I learned some things. It is certainly an important and under-studied subject. It was relatively well-written, but there were still some very obvious problems with the English language and the grammar. Proof-reading by a native English speaker would improve the draft substantially. It will be a good and important article with some corrections and modifications. One major problem is a large number of conclusions without citations to supporting references. I provided a lot of feedback below, which should help you get this draft ready for publication.
Page 2, line 78: “Studies where the full text was unavailable were also excluded.” Please provide a little bit more information. Did you request full texts from the authors when the articles were unavailable? If so, did you receive responses?
Table 1: The search strategy is somewhat unclear. For instance, why were different pain and nutrition search words were used for each search term?
Page 3, line 92: “Malnutrition refers to undernutrition (underweight) and over-nutrition (overweight and obesity).” This doesn’t sound quite right. I don’t think undernutrition = underweight and over-nutrition = overweight and obesity. More accurately, undernutrition can lead to a person becoming underweight and over-nutrition can lead to a person becoming overweight or obese.
Page 3, line 97: “(i.e. under-, overweight or obese).” Spell out “underweight” here.
Page 3, line 113: “In 2020, ePPOC reported that 2% of the 20,000 patients were underweight (BMI < 18.5 kg/m2) [17].” Please clarify that this 20,000 patient sample all had chronic pain conditions. Also, the reader would probably like to know what percentage of these patients were overweight.
Page 3, line 120: “… and lead to poor health outcomes such as reduced life expectancy and quality of life.” Please find a reference to support this statement.
Page 3, line 122: “In Australia, up to 50% of older adults (> 65 years) are malnourished or at an increased risk of malnutrition.” Please find a reference to support this statement.
Page 4, line 128: “Frailty is characterised by a decline in physical, mental and multisystem function.” The term “frailty” is used often in the paper. It is a somewhat vague term. Are you referring to physical deconditioning? Please define it more clearly.
Page 4, line 136: “The western lifestyle and diet are contributing factors impacting the global development of overnutrition, or excess weight (overweight and obesity).” Please find a reference to support this statement.
Page 4, line 140: “This trend is also confirmed in the clinical populations.” Change this sentence to “This trend has also been confirmed in clinical populations.”
Page 4, line 144: “Undernutrition, on the other side,….” Change this to “Undernutrition, on the other hand,….”
Page 4, line 147: “…for example, higher intensity,…” I assume you mean pain intensity. If so, please change to “…for example, higher pain intensity,…”
Page 4, line 147: “A summary of common pain locations (local pain) related to over-, and undernutrition is showed in Figure 1 except from multiple sites pain or spreading pain including regional (e.g., myofascial pain syndrome) and widespread pain (e.g., fibromyalgia).” This is a run-on sentence. Please shorten it or break it into two sentences.
Figure 1: I like this figure, but it needs a lot more explanation. How are each of these pain sites related to poor nutrition? You could consider placing the figure later in the draft, after you have provided some rationale in the text for the relationship between nutrition and these pain sites.
Page 5, line 160: “Findings to date provide the most prevalent causes of chronic pain diagnosis related to nutritional status and diet intervention include….” It is not adequate to say that all these pain conditions are related to nutritional status without providing more information. How was “nutritional status” determined? In what way was it demonstrated that nutritional status was associated with pain and pain-related symptoms? You also need to provide more information about “diet intervention” as it relates to these disorders.
Page 5, line 168: “Poor nutrition does not have an isolated impact on chronic pain pathophysiology but also worsens comorbidities related to chronic pain. For instance, studies on chronic pain patients with obesity demonstrated the outcomes such as physical limitation [54,55], psychological wellbeing [56], sleep disturbances [57,58], poor health-related quality of life (HRQoL) [59,60], and function dependence [61,62].” These sentences are awkward. You message isn’t clear. Please rewrite them.
Title 3.2 “Altered Eating Behaviour and Dietary Preferences” and Page 5, line 185: “… it is clear that chronic pain patients may have altered eating behaviour.” Is “altered” is what you mean here. Altered means that it has changed. You have provided research evidence that chronic pain patients may be more likely to eat poorly, but how do you know their diets have changed since their chronic pain conditions began? Consider changing the title to “3.2. Eating Behaviour and Dietary Factors in Subjects with Chronic Pain” and the sentence to “… it is clear that some chronic pain patients may have poor dietary habits” or something similar.
Page 5, line 190: “It is also suggested that people experiencing pain generally consume more energy, added sugars, saturated fatty acids, sodium, and caffeine.” What does energy mean in this context? Same with the next few sentences. If you mean calories, then please replace the word “energy” with “calories.”
Page 5, line 202: “Additionally, for patients with undernutrition, their suffering could be accompanied by loss of appetite and decreased food intake. This leads to poor dietary intake or absorption of nutrients and subsequently decreased fat free mass and impaired functions.” Change to “Additionally, for patients with undernutrition, their pain-related suffering could be accompanied by ….” And “This could lead to …..” Also, you will need to explain how decreased food intake can lead to poor absorption of nutrients. You need to define “impaired functions.”
Page 6, line 202: “Identifying and understanding these mechanisms subsequently increases the effectiveness of nutrition assessments and …” Change to “Identifying and understanding these mechanisms can potentially increase the effectiveness of nutrition assessments and …”
Page 6, line 219: “species?” Is this the correct word here?
I stopped commenting on the English language and grammar here. There are still multiple problems in the rest of the draft that need to be corrected.
Page 6, line 227: “Inversely, increased dietary antioxidant intake and increased detoxifying ability of the body can alleviate pain among chronic musculoskeletal pain population.” This is a very strong statement, based on a small study with 17 subjects. You need to temper your conclusions here, like “There is some evidence that increased dietary antioxidant intake and increased detoxifying ability of the body can alleviate pain among chronic musculoskeletal pain population.”
Page 7, line 242: “This finding is supported by a number of pain studies on inflammatory biomarkers, released by the immune cells in response to inflammatory stimulations, including cytokines and chemokines, evokes pain via stimulation of nociceptors, changes in neuronal structures, and sensitization of peripheral and central nervous system [81,83].” This is a run-on sentence that needs to be cut into 2 or three separate sentences.
Page 7, line 268: “Thus, the microbiota-gut-brain axis has been defined as a target for nutritional interventions.” “Defined” is not the right word here. Use “identified” instead.
Page 8, line 307: “A decrease in adipose tissue can also improve pain sensitivity in chronic pain population….” You need to cite a supporting reference here. Also, it needs to be “populations” instead of “population.”
Page 9, line 340: “Obesity is considered as an inflammatory state and has been shown as an important risk factor for various metabolic changes and for many chronic diseases including cardiovascular diseases, cancer, diabetes mellitus, chronic pain.” You need to cite a supporting reference here.
Page 9, line 356: “In vivo studies suggest that exposure to a high fat diet for 1 day causes glial cell activations for 2 weeks.” This appears to be a study with rats. You need to specify that. These results may not translate to human subjects.
Page 9, line 373: “Dysregulated epigenetic marks can lead chronic diseases.” You need to cite a supporting reference here.
Page 10, line 418: “Most health services or facilities will nominate a specific screening tool…” “Nominate” is not the correct word here. Replace it with “choose.”
Page 10, line 418: “It is also useful to identify potentially serious illness e.g. cancer.” This sentence is out of place in your discussion of weight and BMI assessment. Looks to me like it would fit better in the next paragraph.
Page 11, line 445: “…many of the nutrition recommendations in “Table 2” will not only improve pain experiences but will also improve the severity and impact of other chronic health conditions.” Change to “…many of the nutrition recommendations in “Table 2” may not only improve pain experiences but may also improve the severity and impact of other chronic health conditions.”
Table 2. Are these published recommendations? You mention “clinical guidelines” in the text but have not cited any references. One suggestion in the “Dairy and alternatives” section. You recommend individual tubs of yoghurt. A lot of those are loaded with sugar. Consider recommending lower sugar yoghurt. In fact, shouldn’t low sugar intake be a general recommendation? Also, you state that “dehydration increases sensitivity to pain.” Maybe I missed it, but I could find no rationale or supporting citations for a relationship between dehydration and increased pain sensitivity. You either need to provide some support for this claim or delete it.
Page 11, line 449: “Identifying abnormal biochemistry results.” This is an incomplete sentence. Is it supposed to be a title for a new section of the manuscript? In fact, I see that an incomplete sentence begins the next few paragraphs. If these are titles for different sections, then format them that way.
Page 11, line 449: “…there are several micronutrient deficiencies that are commonly associated with chronic pain such as B-450 group vitamins and Vitamin D.” When you make a statement like this, you need to provide a supporting reference. In fact, there are a number of statements in this paragraph that are lacking supporting references.
Page 11, line 482: “Unhealthy eating can be defined as a limited intake of core nutrient-rich foods and excess intake of energy-dense nutrient-poor foods.” Shouldn’t this definition appear near the beginning of the manuscript?
Page 15, line 606: “Current dietary guidelines have been developed for a healthy population.” You need to cite a reference here.
Page 15, line 638: “….emerging evidence that poor nutrition and dietary intake plays a key role in the development and management of chronic pain.” Again, you need a supporting reference. You can use the Brain et al 2019 reference, or others. Now, regarding this statement, you have provided evidence of a correlational relationship between nutritional factors and chronic pain. However, I question whether you have provided specific evidence of a causal relationship - that diet specifically contributes to the development of chronic pain.
Round 2
Reviewer 2 Report
Good job with your edits. The paper flows nicely. You provided good rationale for each of your points. I see lots of supporting references. I have just a few suggestions below. Well done!
Page 9 lines 199 to 204: “For example, it was found that over 25% of chronic pain patients had a comorbidity of obesity [43], much higher than the general population in Sweden [44]. The prevalence was even higher in Australia. , uUp to 45% patients from a tertiary pain clinic waswere classed as obese [45].
Page 15 line 321: remove the comma from this sentence: “ Neuroinflammation, is a localized inflammatory response that occurs in the peripheral and central nervous system [91].”
Page 18 line 380: “In another study, people with chronic musculoskeletal pain who followed a low carbohydrate diet and had a decrease in serum inflammatory biomarkers and pain sensitivity [107].”
Page 26 line 578: Take another look at this sentence. It doesn’t look quite right. “Some of these strategies such as the elimination component of the low FODMAP diet, are nutritional inadequate [162]….”
Page 28 line 617: “Ultra-processed foods undergo several industrial food processes and contain high amounts of sugar, protein and oil products and food additives [170].” I was confused by the inclusion of “protein and oil products.” I would consider protein to be a good thing. I was under the impression that most processed foods contained very limited proteins. I’m not sure what you mean by “oil products” and why they would be bad.
Page 30 line 646: “Food insecurity is the inability (not ability) to reliably access adequate and affordable nutritious food and it is associated with chronic pain and poor mental health.”
Page 30 line 646: “Regardless of a patient’s (not patients’) needs, whether…)”
Lastly, you mention studies on the negative health effects of “high fat” diets. However, you also mention the benefits of good fats and keto diets (which are high in fat). If possible (depending on the studies), you should consider saying “high saturated fat” diets instead.
